# Reducing Sample Size While Improving Equity in Vaccine Clinical Trials: A Machine Learning-Based Recruitment Methodology with Application to Improving Trials of Hepatitis C Virus Vaccines in People Who Inject Drugs

**DOI:** 10.3390/healthcare12060644

**Published:** 2024-03-13

**Authors:** Richard Chiu, Eric Tatara, Mary Ellen Mackesy-Amiti, Kimberly Page, Jonathan Ozik, Basmattee Boodram, Harel Dahari, Alexander Gutfraind

**Affiliations:** 1Department of Medicine, University of Illinois College of Medicine at Chicago, Chicago, IL 60612, USA; rchiu8@uic.edu; 2The Program for Experimental & Theoretical Modeling, Department of Medicine, Division of Hepatology, Stritch School of Medicine, Loyola University Chicago, Maywood, IL 60660, USA; hdahari@luc.edu; 3Consortium for Advanced Science and Engineering, University of Chicago, Chicago, IL 60637, USA; 4Argonne National Laboratory, Lemont, IL 60439, USA; 5Division of Community Health Sciences, School of Public Health, University of Illinois at Chicago, Chicago, IL 60612, USAbboodram@uic.edu (B.B.); 6Department of Internal Medicine, Division of Epidemiology, Biostatistics and Preventive Medicine, University of New Mexico Health Sciences Center, Albuquerque, NM 87131, USA; pagek@salud.unm.edu

**Keywords:** randomized clinical trial, vaccine trial recruitment, hepatitis C, equity, people who inject drugs, machine learning

## Abstract

Despite the availability of direct-acting antivirals that cure individuals infected with the hepatitis C virus (HCV), developing a vaccine is critically needed in achieving HCV elimination. HCV vaccine trials have been performed in populations with high incidence of new HCV infection such as people who inject drugs (PWID). Developing strategies of optimal recruitment of PWID for HCV vaccine trials could reduce sample size, follow-up costs and disparities in enrollment. We investigate trial recruitment informed by machine learning and evaluate a strategy for HCV vaccine trials termed PREDICTEE—Predictive Recruitment and Enrichment method balancing Demographics and Incidence for Clinical Trial Equity and Efficiency. PREDICTEE utilizes a survival analysis model applied to trial candidates, considering their demographic and injection characteristics to predict the candidate’s probability of HCV infection during the trial. The decision to recruit considers both the candidate’s predicted incidence and demographic characteristics such as age, sex, and race. We evaluated PREDICTEE using in silico methods, in which we first generated a synthetic candidate pool and their respective HCV infection events using HepCEP, a validated agent-based simulation model of HCV transmission among PWID in metropolitan Chicago. We then compared PREDICTEE to conventional recruitment of high-risk PWID who share drugs or injection equipment in terms of sample size and recruitment equity, with the latter measured by participation-to-prevalence ratio (PPR) across age, sex, and race. Comparing conventional recruitment to PREDICTEE found a reduction in sample size from 802 (95%: 642–1010) to 278 (95%: 264–294) with PREDICTEE, while also reducing screening requirements by 30%. Simultaneously, PPR increased from 0.475 (95%: 0.356–0.568) to 0.754 (95%: 0.685–0.834). Even when targeting a dissimilar maximally balanced population in which achieving recruitment equity would be more difficult, PREDICTEE is able to reduce sample size from 802 (95%: 642–1010) to 304 (95%: 288–322) while improving PPR to 0.807 (95%: 0.792–0.821). PREDICTEE presents a promising strategy for HCV clinical trial recruitment, achieving sample size reduction while improving recruitment equity.

## 1. Introduction

Despite over thirty years of research, there is still no vaccine for hepatitis C virus (HCV) infection [1,2], a virus affecting over 2 million adults in the U.S. alone [3,4]. While HCV can be cured with direct-acting antivirals (DAAs) [5], treatment with DAAs can cost upwards of $25,000 [6] and does not prevent reinfection [7]. A vaccine promises to be an inexpensive, feasible, and accessible viral control strategy, especially for high-risk populations such as people who inject drugs (PWID), who are affected by the opioid crisis [8].

Our primary motivation for improving vaccine clinical trial recruitment is to reduce the sample size needed in an HCV vaccine trial to achieve adequate statistical power. An analysis of randomized clinical trials (RCTs) revealed that 19% were terminated due to insufficient accrual of participants [9], and up to 86% do not reach their recruitment targets within the originally envisaged timescale [10,11]. Moreover, studies place the cost of recruitment to a vaccine clinical trial to be in the range of $9000 to $16,000 per recruited subject, considering expenses such as tracking and retention costs, reimbursement, and drug costs [12,13]. Reducing sample size therefore not only increases the feasibility of vaccine trials, but also enables significant cost savings.

We secondarily seek to improve representation of diverse populations in HCV RCTs and ensure equitable access to novel vaccines. A study by Wilder et al. found that participation of Black/African American participants in HCV-related clinical trials does not proportionally reflect the burden of HCV among this demographic in North America [14]. The issue of adequate representation is particularly pertinent in the context of HCV because demographic characteristics such as sex [15,16,17], race [15,18], and age [19] can impact HCV viral clearance and immunology and likely alter vaccine efficacy. The FDA recently called for innovative strategies promoting enrollment of underrepresented populations in RCTs [20] and the NIH instructed trials it funds to ensure adequate inclusion of minorities [21]. However, it is desired to achieve these objectives without magnifying the costs of screening and the overall sample size.

This study aims to address both sample size and equity concerns by introducing a novel approach for recruiting for HCV vaccine clinical trials. The approach integrates machine learning (ML) to identify persons most likely to be exposed to HCV while also implementing a weighting scheme prioritizing underrepresented candidates. We refer to this recruitment strategy as the Predictive Recruitment and Enrichment method balancing Demographics and Incidence for Clinical Trial Equity and Efficiency (PREDICTEE). We surveyed the literature for similar work and identified several studies that used a predictive model for recruitment of individuals at high risk for the outcome of interest [22,23,24,25] as well as others that discuss strategies to improve equity and inclusion in RCTs [26]. There have also been other suggestions for improving trials for HCV vaccine candidates (e.g., [27,28]), but none targeting sample size or equity. Therefore, we believe this study is the first to achieve the two opposing objectives in the context of vaccine trial recruitment for HCV vaccines.

This study contributes to a growing body of literature on the implications of artificial intelligence (AI) and machine learning in clinical trials. Recent reviews by Ismail et al. [29] and Harrer et al. [30] describe how AI/ML may be leveraged to optimize recruitment to reduce trial failures, optimize patient composition, and reduce time and costs associated with conducting an RCT. We believe PREDICTEE offers one such ML-based solution to these goals. This study also complements a recent study by Oikonomou et al. [31] which describes a ML-based phenomapping strategy capable of maximizing RCT enrollment efficiency that is similar to what we propose in our PREDICTEE methodology. However, we offer an alternative method of optimizing the clinical trial cohort and apply it to a simulated vaccine clinical trial.

PREDICTEE also builds upon our previous work on a model of Hepatitis C elimination in PWID (HepCEP), which simulates HCV infection, network formation, and syringe sharing in the PWID population of metropolitan Chicago [32]. The present study investigates how longitudinal PWID data, such as that generated using the HepCEP model, can be leveraged to improve vaccine trial recruitment equity and efficiency among PWID. We also propose a novel use of survival analysis in clinical trial recruitment decisions for the purpose of prognostic enrichment. Using simulations of HCV vaccine trial recruitment of PWID based on data collected from our previous study, we show that PREDICTEE yields trial cohorts that are approximately half the size of those recruited using conventional methods, while also improving demographic representativeness.

## 2. Materials and Methods

In designing PREDICTEE, we set multiple objectives: (1) to identify subgroups with high incidence in the trial cohort while ensuring that (2) the final sample satisfies constraints on representation. Additionally, the recruitment process should (3) be efficient and conclude in a pre-specified amount of time and (4) cope with uncertainty about the quality and background of trial candidates. Below, we describe how PREDICTEE works and evaluate its performance in a simulated HCV vaccine RCT in Chicago.

### 2.1. Preparation

#### 2.1.1. Longitudinal Data

Our longitudinal data on PWID are derived from a large synthetic PWID population. To achieve this, we used the HepCEP agent-based model that simulates PWID behavioral patterns–daily injection drug use, social network formation and dissolution, and geography [32,33]. Details of the HepCEP model are described in the Appendix A; in brief, the HepCEP model simulates events such as PWID attrition, new PWID arrival, drug sharing, network formation, HCV infection, recovery, vaccination and more. To generate our synthetic population, we constrained the HepCEP model to maintain a population of 100,000 PWID over its 10-year simulation, removed any simulated vaccine effects, and kept all other HepCEP parameters at default. This generated profiles for 123,071 PWID—the pool of candidates we used for our vaccine trial recruitment. Each PWID profile contains demographic, network, and injection characteristics, as well as the recorded time to HCV infection if it occurred. Additional variables were also calculated for these synthetic PWID using the simulated HepCEP events including an indicator variable denoting if the individual would become infected in a clinical trial with a 1.5-year follow-up period and if the individual is HCV-susceptible (HCV RNA/Ab-negative).

#### 2.1.2. Survival Analysis Models

To achieve its goals of minimizing sample size, PREDICTEE relies on a survival model that estimates the probability of HCV infection by the end of the clinical trial follow-up period, given the demographic, network, and injection characteristics of individual PWID. This trained survival model will be used in the PREDICTEE process to enrich the trial cohort with high-risk PWID most likely to experience acute HCV infection, increasing the cohort incidence of HCV and thereby decreasing the required sample size needed to attain adequate power. In our simulated HCV vaccine trials, we tested two models: (1) a Cox proportional hazards model, a classic survival analysis model [34], and (2) a non-parametric random survival forest (RSF), a more recently developed ensemble prediction method [35]. For each simulation, we implemented a 20/80 train-test split, in which 20% of the synthetic population of 123,071 PWID was used to train the survival analysis model, and the remaining 80% served as the recruitment pool from which simulated trial participants were recruited from. Both Cox and RSF models were trained using the demographic, behavioral, and network attributes provided by the HepCEP model, listed as follows:Age;Sex assigned at birth;Syringe source (harm reduction program or other);The number of PWID who gave the candidate drugs/injection equipment in the last 30 days;The number of PWID who the candidate gave drugs/injection equipment to in the last 30 days;The total number of people in the candidate’s drug use network;The number of daily injections;The fraction of injections that involve receiving drugs or injection equipment from another person in the network.

The outcome variable used to train the models was the time (in years) until the PWID experience an acute HCV infection. Any PWID who did not experience an HCV infection during the 10-year HepCEP simulation were right-censored at 10 years. These trained Cox and RSF models are used to predict the probability that new PWID will be infected with HCV by 1.5 years after enrollment, allowing for the recruitment of high-risk PWID to maximize trial cohort incidence.

The RSF model also requires the selection of two main hyperparameters: number of trees (ntree) and number of variables randomly selected as candidates for splitting a node (mtry). Using the recommendations in the literature [36,37], we set mtry = *p*/3, with *p* being the number of explanatory variables; thus, mtry = 3 because there are eight explanatory variables used to train the model. Additionally, we set ntree = 100 based on [38,39].

#### 2.1.3. Demographic Targets

While PREDICTEE primarily aims to achieve a minimal sample size through PWID risk prediction, it also attempts to improve trial generalizability by recruiting a trial cohort that resembles a prespecified target demographic. We balanced both objectives through a dynamic weighting and scoring process, which factors in both a candidate’s HCV risk as well as the characteristics needed to reach the target demographic composition. Details on the scoring equation are described later in this section.

### 2.2. The PREDICTEE Workflow

An overview of PREDICTEE recruitment as it was simulated in this study is illustrated in Figure 1. PREDICTEE first requires pre-existing longitudinal data of the recruited population, containing characteristics predictive of HCV infection and a timeline of infection events. The synthetic population of 123,071 PWID serves this purpose for our simulations. We used 20% of these data to train a Cox or RSF model, and the remaining 80% served as the recruitment pool. PWID screening was simulated via random sampling of this recruitment pool. Screened candidates undergo immediate scoring, factoring in demographic considerations and their probability of HCV infection, estimated using the survival model. Once a batch of *B* candidates is screened and scored, the candidates with the highest scores receive HCV screening, and the top-scoring *R* HCV-susceptible PWID are recruited and randomized to vaccine or control groups.

After each batch, we update the weights used for scoring candidates based on the composition of the partial cohort. At a predetermined point, we also optionally perform sample size re-estimation based on the predicted incidence of the partial cohort, calculated by averaging the individual infection probabilities of the recruits, and allow for early termination of recruitment. The point at which sample size is re-estimated is inversely proportional to the estimated incidence improvement of the predictive model compared to conventional recruitment methods. For example, a model that is expected to double the incidence in the trial cohort would re-estimate sample size when half the required sample size of conventional recruitment is reached. These parameters would be determined via simulations using the same preliminary data that were used to train the predictive model. For a list of parameters used in PREDICTEE, see the Appendix A. In subsequent sections, we describe the processes of weighting and scoring in greater detail.

#### 2.2.1. Candidate Scoring Equations

The arriving candidates are denoted $c_{1}$, $c_{2}$, …, and a new candidate $c_{i}$ arrives at a time point *t*, where *t* represents the number of prior batches that have already been considered. The score ${Score}_{t}(c_{i})$ is based on several factors: (1) the cohort of previously recruited subjects, denoted $A_{t-1}$, with individual recruited candidates denoted *a*; (2) the target composition for the final cohort across each population category (e.g., proportion of non-Hispanic Black), denoted $p_{j}$, where j∈J, with *J* representing the larger set of all considered categories (demographic and/or others); and (3) the characteristics of candidate $c_{i}$. The latter includes the candidate’s predicted incidence of HCV, denoted $I(c_{i})$, as estimated by the predictive model, and the candidate’s categories, denoted $d_{j}(c_{i})$, which equal 1 when the person is a member of category j and 0 otherwise. The candidate score is given by Equation (1):(1)$${Score}_{t}(c_{i})=w_{t}I(c_{i})+(100-w_{t})\sum_{j\in J}d_{j}(c_{i})\left[p_{j}-\frac{1}{\left|A_{t-1}\right|}\sum_{a\in A_{t-1}}d_{j}(a)\right]$$
where $w_{t}$ is the weight assigned to incidence and ${100-w}_{t}$ is the weight of categorical factors. This scoring function simultaneously addresses the conflicting goals of trial equity while decreasing sample size via recruiting higher-risk candidates. For an example of candidate scoring, see Appendix A.

After score assignment, the top-*R*-scoring HCV-naïve candidates are recruited. Because we test for susceptibility to HCV after scoring, if any of these *R* candidates are not HCV-susceptible, they are replaced by the next-best-scoring naïve candidate. A trial could optionally maintain a backlog of candidates who were not recruited and consider them alongside subsequent batches, which would help decrease batch variability and maintain a steady recruitment screening, particularly in situations when the value of *B* is small. This process is repeated until the end sample size is reached. The values of *B* and *R* can be altered based on trial capabilities. The ratio between them, *R/B*, is the proportion of each batch that is recruited. Decreasing the ratio minimizes the size of the sample, while increasing it minimizes the cost of screening.

Incidence and demographic representativeness are weighted dynamically, with PREDICTEE prioritizing incidence at the beginning of the trial and gradually giving more weight to closing population gaps as the trial advances. Thus, the initial $w_{0}$ is always 100, and then it decreases after each batch according to Equation (2):(2)$$w_{t}=w_{t-1}-\frac{R}{N}(w_{t-1}-w_{min})$$
where *w_t_* is the incidence weight for the current batch *t*, *w_t_*_−1_ is the incidence weight for the last batch *t*−1, $w_{min}$ is the incidence weight floor, *R* is the number of candidates recruited per batch, and *N* is the most recent estimated sample size. For simulations where we anticipate a large gap from targets, we accelerated the rate of decrease, as described in the Appendix A. This weighting approach would allow investigators to first establish a baseline profile of recruited candidates at the beginning of a trial based on the highest-risk candidates screened. As the trial progresses, this profile of candidates can be leveraged to address sampling inequities without significantly compromising the main objective of maintaining a low sample size.

Optionally, the trial plan could allow for early stopping of recruitment based on the cohort incidence. Namely, the stopping decision could consider the expected number of events in the recruited cohort as estimated by the model, without unblinding the arms or waiting to observe actual events. It is recommended that the decision threshold include a margin of safety for any model error.

### 2.3. Evaluating PREDICTEE HCV Vaccine Trial in Chicago PWID Population

#### 2.3.1. Design of Simulation Experiments

In previous work, we simulated trials of HCV vaccines end-to-end [32,33,40,41]. In this work, we implement software that simulates only the outcome of the recruitment process. We applied PREDICTEE to two challenges in building a representative cohort: (a) a study cohort matching the Chicago area’s PWID population in terms of race/ethnicity, sex, and age distribution; (b) a maximally balanced cohort with a racial composition target for Hispanic, non-Hispanic Black (NHBlack), non-Hispanic White (NHWhite), and Other individuals of 33:33:33:1, respectively, and a sex composition of 50:50 (male–female). For simulations in which age was targeted, each candidate was categorized into ten-year age groups. Age was not considered in the maximally balanced cohort (Scenario b) because of the extremely low prevalence (~3.5% of the recruitment pool) and HCV incidence in candidates in the 49+ age group in our Chicago PWID data. This would make it difficult to match a balanced cohort without arbitrarily decreasing the target for candidates in the 49+ group. To compare PREDICTEE to conventional recruitment strategies, we also simulated two additional recruitment strategies using the Chicago area’s PWID dataset, as follows. (1) Random/uniform: candidates are an unbiased random sample from the HCV-susceptible PWID population; (2) in-network: only candidates who receive syringes from their social network are recruited because they have a higher incidence rate (as proposed in [32,33,42]). Random recruitment is offered as a synthetic benchmark, although few trials can realistically implement the random recruitment of PWID due to the complexity of working with the PWID population, and it is avoided in sites like Chicago due to the low incidence rate.

For both Cox and RSF models, PREDICTEE was executed 100 times for each of 100 random train/test splits of the synthetic population, yielding 10,000 total trial recruitment simulations. For each simulation, we calculate Harrell’s concordance index (C-index) for the unique trained model. The C-index is a goodness-of-fit measurement commonly used for survival analysis models with censored data, analogous to the area under the ROC curve (AUC) for more classic predictive models and diagnostic tests [43,44]. Based on the existing literature, the C-index for a survival analysis model should be at least 0.7 to adequately discriminate between risk profiles [45,46,47]. For all simulation runs, assumed vaccine efficacy was 60% and the trial follow-up period was 1.5 years, following the work of Page et al. [2]. We define vaccine efficacy as the proportionate reduction in chronic cases between the unvaccinated and vaccinated groups [48]. The initial required sample size was set to 800 to achieve 80% power based on the estimated 1.5-year cumulative incidence of 5.5% from the in-network recruitment method. Parameters were set accordingly: *B* = 50, *R* = 5, and $w_{min}=25$. The candidate arrival process was simulated via random sampling from the recruitment pool. Simulations of random sampling and in-network recruitment were also repeated 10,000 times.

Sample size re-estimation occurred after 400 candidates recruited for Cox PREDICTEE and 267 candidates recruited for RSF PREDICTEE based on experimental data that showed a required sample size of approximately 800 candidates for in-network recruitment and that the Cox and RSF model would approximately double and triple the incidence, respectively. At this re-estimation point, the predicted incidence of the partial cohort is used to update the sample size rather than looking at the observed outcomes of recruits. This is achieved by averaging the infection probabilities generated by the Cox or RSF models for all candidates recruited to the trial up to the re-estimation point, without revealing treatment assignments. Due to the blinded nature of this process, there is a minimal impact on type I error [49]. Specific parameters and formulae involved in sample size re-estimation can be found in the Appendix A.

Incidence values in this study represent cumulative incidence over the 1.5-year trial follow-up period, expressed as the proportion of recruited participants who develop HCV. Predicted incidence in PREDICTEE simulations is compared to observed incidence to confirm model validity (see the Appendix A). Demographic statistics are recorded for each trial cohort and averaged across all runs for both Cox and RSF PREDICTEE. These are plotted alongside the target population and in-network recruitment to evaluate PREDICTEE’s demographic adjustment capability.

For each recruitment method and demographic category, we also calculate participation-to-prevalence ratio (PPR), a metric that evaluates demographic representation and has been widely used in clinical trial settings for assessing adequate enrollment of demographic subgroups [50,51,52,53,54]. PPR = proportion among trial participants/proportion among disease population. Thus, a PPR of 1.0 for a specific demographic represents perfect representation. A PPR of less than 0.8 signifies underrepresentation, a PPR between 0.8 and 1.2 signifies adequate representation, and a PPR greater than 1.2 signifies overrepresentation. For our simulations, we report the average of the lowest PPR across all target categories in each run, denoted PPR_min_.

#### 2.3.2. Software Modeling Platform and Tools

Trial power analysis and simulations were conducted using R, with Cox and RSF models trained using the survival and randomForestSRC packages, respectively [55,56,57].

## 3. Results

Our analysis below compares recruitment techniques in terms of required sample size, representativeness, and screening requirements. We also compared PREDICTEE using the Cox model against RSF.

### 3.1. Matching the Chicago PWID Population

The baseline characteristics of the Chicago PWID population are listed in Table 1. Unless otherwise specified, PREDICTEE attempts to match the HCV-susceptible PWID population (third column)—this is the population most likely to eventually receive the vaccine. In-network recruitment represents a good method for recruitment into HCV trials without a predictive model and serves as a comparison for PREDICTEE. As can be seen in Table 1, in-network recruitment results in a significant underrepresentation of non-Hispanic Black candidates compared to the HCV-susceptible population.

PREDICTEE led to a marked increase in cumulative incidence over the course of a simulated trial, as shown in Table 2a. Application of the Cox model led to a nearly two-fold increase in incidence compared to in-network recruitment from 0.055 (95%: 0.044–0.068) to 0.097 (95%: 0.090–0.104), and the RSF model led to a nearly three-fold increase to 0.149 (95%: 0.141–0.155). This corresponds to a sample size of 444 and 278 for Cox and RSF PREDICTEE, respectively, compared to 802 for in-network recruitment (a ratio of 1.81 and 2.88, respectively). In terms of screening requirements, Cox PREDICTEE achieved a smaller sample size with an approximate 12% increase in eligibility screening; however, RSF PREDICTEE achieved its reduction in sample size while also reducing screening requirements by almost 30%.

While improving incidence, PREDICTEE simultaneously corrected for deviations in demographic representativeness seen in in-network recruitment, illustrated by Figure 2A. This is also expressed in the PPR_min_ row in Table 2a, which shows that PPR_min_ increased greatly from 0.475 (95%: 0.356–0.568) with in-network recruitment to 0.764 (95%: 0.593–0.934) for Cox and 0.754 (95%: 0.685–0.834) for RSF. It should be noted that PPR_min_ remained under 0.80 for PREDICTEE recruitment exclusively because of the over-49 age group that is being matched. This is due to the low PWID prevalence and HCV incidence of this demographic, leading to continued underrepresentation. If this demographic were not considered in matching, PPR_min_ was calculated to be 0.947 (95%: 0.924–0.970) for Cox PREDICTEE and 0.964 (95%: 0.948–0.974) for RSF PREDICTEE.

### 3.2. Targeting Arbitrary Demographics

Results for matching a maximally balanced population to assess the ability of PREDICTEE to adjust to more dissimilar targets are summarized in Table 2b and Figure 2B. In this scenario, in-network recruitment expressed significant inequities with a PPR_min_ of 0.250 (95%: 0.221–0.280), while PREDICTEE was able to close demographic gaps reasonably well, with an average PPR_min_ across the simulations of 0.903 (95%: 0.889–0.918) and 0.807 (95%: 0.792–0.821) for Cox and RSF PREDICTEE, respectively. At the same time, PREDICTEE increased cumulative incidence to 0.085 (95%: 0.079–0.095) and 0.137 (95%: 0.130–0.144), respectively. This corresponds to a required sample size of 506 and 304 for Cox and RSF PREDICTEE compared to 802 for in-network recruitment (a ratio of 1.58 and 2.64, respectively). Cox PREDICTEE achieved a smaller sample size with an approximately 28% increase in eligibility screening, while RSF PREDICTEE achieved its reduction in sample size while also reducing screening requirements by roughly 23% compared to in-network recruitment.

The models were generally quite accurate in predicting time to infection in the simulated epidemic; across 10,000 simulations, the average C-index value was 0.871 for Cox models and 0.946 for RSF models, indicating adequate risk profile discrimination.

## 4. Discussion

Our study developed PREDICTEE as a proposed method for reducing the required sample size within a vaccine clinical trial while ensuring cohort representativeness to a pre-specified target population. We have shown that contemporary techniques in recruitment may result in demographic differences within the trial cohort (Table 1), which may contribute to a lack of generalizability if these characteristics are relevant to vaccine efficacy or virus epidemiology. Through simulation, PREDICTEE illustrated its ability to decrease sample size two- to three-fold while also improving recruitment equity of target characteristics compared to conventional recruitment. It achieved this by selectively recruiting high-incidence PWID while prioritizing underrepresented demographics (Table 2a and Figure 2A). This was the case even in circumstances with a highly dissimilar target population (Table 2b and Figure 2B).

### 4.1. Implications

PREDICTEE complements and contributes to the literature on trial design strategies that leverage predictive models. Previous work examined ideas such as prognostic enrichment, adaptive designs, and sample size re-estimation. The statistical and clinical benefits of these strategies have been well documented, allowing trials to not only improve their efficiency and impact but also to ensure adequate statistical power [58,59]. We show in a simulated setting that PREDICTEE is able to maintain the benefits that adaptive trial designs offer while also providing further benefits to clinical trial recruitment through the implementation of both a predictive model and a weighting system prioritizing underrecruited categories of candidates.

A practical drawback of previous adaptive enrichment designs is that patient recruitment may be halted before interim analysis so that primary outcomes can be assessed, increasing trial duration and delaying submission of experimental agents for approval [60,61]. PREDICTEE presents a potential solution to this as it offers a predicted incidence value. Sample size could be re-estimated without unblinding, using only the data recorded at enrollment and before any events are observed. As a result, clinical trials would not need to pause recruitment. Naturally, this scheme relies on a well-calibrated predictive model and may necessitate a margin for any model error.

We also consider the ability of PREDICTEE to serve as an alternative to existing estimators that aim to generalize the results of clinical trials to target populations by accounting for nonrandom sampling [62,63]. By ensuring concordance between desired characteristics such as demographics between the trial cohort and the target population, PREDICTEE essentially aims to generalize trial results via equitable recruitment processes instead of post hoc quantitative methods that may be non-robust to model misspecification or the misestimation of parameters [64].

### 4.2. Limitations

Our quantitative results are based on agent-based simulations of a PWID population from a large metropolitan city. Although these simulations have been shown to be more realistic than aggregate models [65], they are still a simplification of human behavior and network formation. As a result, the survival models might be accurate in the simulation setting but experience increased prediction error in a real clinical trial environment. However, we believe that these simplifications are unlikely to affect the overall conclusions of this study, since simulations were only used to train the predictive models and evaluate the PREDICTEE workflow. The candidate data and recruitment pool that was used in these simulations were derived from real survey data collected from PWID in Chicago (see the Appendix A). As in conventional trials, trial organizers may incorporate a margin of error into their sample size calculations to reduce the likelihood of an underpowered trial.

In designing our PREDICTEE recruitment simulations, we aimed to reflect real-world conditions, but some effects were not captured. Due to a lack of appropriate data on the ease of recruitment and network referrals between PWID, we formed batches for PREDICTEE by randomly sampling from a predefined recruitment pool. Actual feeder processes for RCTs likely observe clustering and uneven sampling due to differences in accessibility of PWID [66,67], but we do not anticipate this having a tangible effect on outcomes. Additionally, loss to follow-up is higher in select subgroups (e.g., those engaged in higher risk activities) [68,69,70], but given the appropriate data, these effects could be easily incorporated into PREDICTEE. The PREDICTEE scheme also increases the HCV incidence in underrepresented demographics to a smaller extent than other demographics; however, this did not have a significant effect in our study (see the Appendix A).

While our simulation results are based on metropolitan PWID data from the Chicago area, we anticipate that PREDICTEE will work in other geographic areas with similar HCV epidemics. However, each site has unique HCV risk factors, and ideally, users should train their own site-specific models. Similarly, all parameters of PREDICTEE (see the Appendix A) should be designed based on local knowledge of the trial site. PREDICTEE would ideally rely on ample and accurate longitudinal local data of the target population to achieve optimal results. From a global lens, this study heavily incorporated a North American context through the use of ethno-racial categories that are prominent in the U.S.; however, PREDICTEE could be applied to a range of categories of interest such as genetic variants, socioeconomic categories, and more.

Finally, more research is needed on the cost-effectiveness of PREDICTEE in comparison to traditional recruitment methods. Although we anticipate that cost savings associated with recruiting fewer subjects would outweigh any additional screening costs with PREDICTEE, other costs would also have to be considered such as predictive model training, algorithm implementation, and personnel costs. This would require further studies on specific real-world PREDICTEE parameters and how they compare to traditional recruitment, such as total clinical trial duration, required volume of pre-trial data, and ease of identifying high-risk PWID.

### 4.3. Extensions

Future research with PREDICTEE should focus on refining the methodology in field conditions and validating its use in other types of geographical regions (e.g., rural). We discuss some possible real-world implementations of PREDICTEE in the Appendix A. The PREDICTEE strategy could also incorporate extensions in which the arms within the trial are balanced both in terms of static characteristics, such as race and sex, as well as the number of predicted infectious events. Imbalance in post-randomization events [71] has been identified as an important source of trial failure in HCV vaccines [41], and it could be reduced by using the survival model to balance the trial arms. Efficiency of recruitment could also be enhanced by using predictive models beyond predicting incidence to, e.g., predict viral status, refusal probability, and/or nonadherence, among others, as demonstrated in a recent COVID-19 vaccine study [72]. The efficiency of predictive recruitment could benefit from schemes for dynamic model retraining, allowing trials to be carried out in sites with limited previous incidence data. Moreover, the adoption of PREDICTEE could be accelerated by the sharing of PWID and HCV epidemiological data via an application programming interface (API). While this study illustrated the benefits of PREDICTEE in the context of an HCV vaccine trial, this method could be easily extended to trials of other vaccines, such as COVID-19 or HIV, or trials for novel treatments that also often seek high-incidence candidates. Moreover, we envision that PREDICTEE could be adapted to trials with non-binary outcomes, such as weight, A1c, etc.

## 5. Conclusions

PREDICTEE is the first recruitment method implementing a predictive model that balances incidence and population representation with the goal of producing more equitable and feasible clinical trials. Our results illustrated that PREDICTEE can recruit high-incidence participants while adjusting recruitment to multiple target populations. Further research is warranted to validate PREDICTEE in field conditions and using a variety of trial designs.

## Figures and Tables

**Figure 1 healthcare-12-00644-f001:**
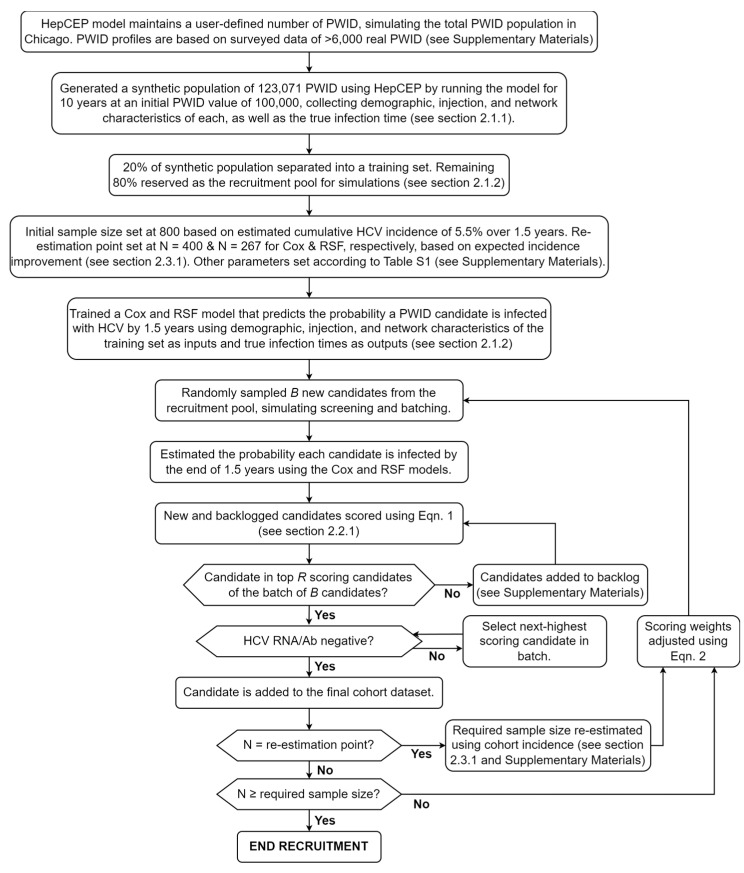
Schematic of the PREDICTEE recruitment method as it was performed in our simulations. During the setup, a predictive model is trained using existing longitudinal data, and parameters for PREDICTEE are set. Recruitment proceeds in cycles of scoring, batching, recruiting, and adjusting parameters. Optionally, sample size is re-estimated at a pre-specified number of PWID recruited (re-estimation point).

**Figure 2 healthcare-12-00644-f002:**
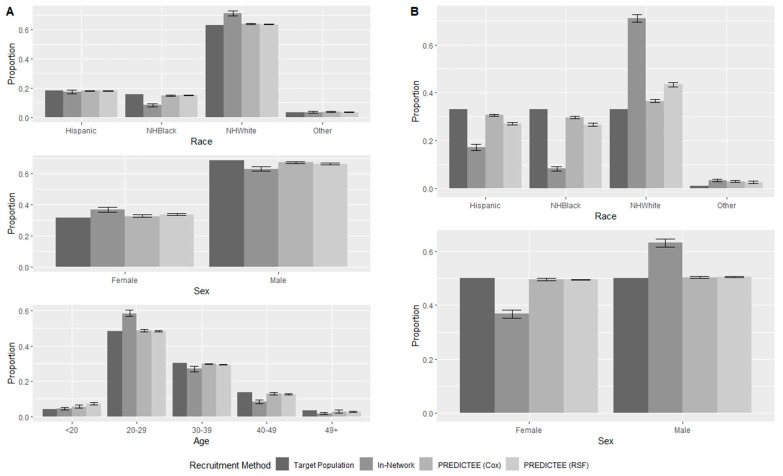
Demographic outcome of recruitment under two different target populations: (**A**) Chicago’s susceptible PWID and (**B**) an arbitrary balanced population. Error bars represent the 95% range over 10,000 trials. Sex reflects assignment at birth.

**Table 1 healthcare-12-00644-t001:** Demographic, behavioral, and network attributes of PWID populations in Chicago.

Attribute	2018 Chicago PWID Population	Susceptible Population with Receptive Network	HCV-Susceptible Population
*Demographic Attributes*			
Location (by ZIP Code)	City: 45.4%	City: 27.5%	City: 36.5%
	Suburbs: 54.6%	Suburbs: 72.5%	Suburbs: 63.5%
Race/Ethnicity	Hispanic: 18.7%	Hispanic: 17.5%	Hispanic: 18.1%
	NH Black: 20.8%	NH Black: 10.0%	NH Black: 15.5%
	NH White: 57.2%	NH White: 69.3%	NH White: 63.2%
	Other: 3.2%	Other: 3.2%	NH Other: 3.2%
Sex	Female: 30.6%	Female: 37.8%	Female: 31.5%
	Male: 69.4%	Male: 62.2%	Male: 68.5%
Age, mean (IQR)	35.2 (26.0–43.0)	29.8 (24.0–34.0)	31.4 (24.9–37.0)
Elapsed years of injection drug use, mean (IQR)	11.3 (3.4–15.6)	6.3 (2.0–8.6)	7.2 (2.5–9.9)
Enrollment in any harm-reduction (HR) program	HR: 48.4%	HR: 33.1%	HR: 45.5%
	Non-HR: 51.6%	Non-HR: 66.9%	Non-HR: 54.5%
HCV Infection State	Infected (acute or chronic): 28.1%	0%—all susceptible	0%—all susceptible
	Recovered (antibody-positive): 12.3%		
*Behavioral Attributes*			
Daily Drug Injections, mean (IQR)	2.5 (0.9–3.3)	2.6 (0.9–3.6)	2.4 (0.8–3.2)
Probability of Receptible Sharing, mean (IQR)	19.4% (0.0–37.3%)	30.3% (5.0–50.0%)	21.3% (0.0–40.6%)
*Network Attributes*			
In Degree (Receptive Network Size)	0 (no network)–68.0%	0 (no network)–0%	0 (no network)–68.0%
	1–23.5%	1–75.7%	1–23.5%
	≥2–8.5%	≥2–24.3%	≥2–8.5%
Out Degree (Giving Network Size)	0 (no network)–72.0%	0 (no network)–45.0%	0 (no network)–69.4%
	1–20.6%	1–40.5%	1–22.7%
	≥2–7.4%	≥2–14.5%	≥2–7.9%

**Table 2 healthcare-12-00644-t002:** Incidence data when PREDICTEE attempts to match two different demographic compositions: (a) Chicago’s susceptible PWID and (b) an arbitrary balanced population. All values represent the mean of 10,000 simulations, with a 95% range calculated using quantiles.

	Random Sample	In-Network Recruitment	PREDICTEE (Cox Model)	PREDICTEE (RSF Model)
**(a)**	**Matching Chicago’s Susceptible PWID**			
Cohort Incidence	0.024 (0.018–0.033)	0.055 (0.044–0.068)	0.097 (0.090–0.104)	0.149 (0.141–0.155)
Required Sample Size (Calculated Using Cohort Incidence)	1876 (1356–2512)	802 (642–1010)	444 (408–480)	278 (264–294)
Expected Number of Candidates Screened Before Achieving Required Sample Size *	2207 (1595–2955)	4648 (3721–5853)	5224 (4800–5647)	3271 (3106–3459)
Post Hoc Power if 800 Recruited	49.2% (40.5–60.5%)	80.0% (71.5–87.1%)	95.5% (94.2–96.7%)	99.5% (99.3–99.6%)
PPR_min_ **	-	0.475 (0.356–0.568)	0.764 (0.593–0.934)	0.754 (0.685–0.834)
**(b)**	**Matching Arbitrary Balanced Demographics**			
Cohort Incidence	0.024 (0.018–0.033)	0.055 (0.044–0.068)	0.085 (0.079–0.095)	0.137 (0.130–0.144)
Required Sample Size (Calculated Using Cohort Incidence)	1876 (1356–2512)	802 (642–1010)	506 (452–550)	304 (288–322)
Expected Number of Candidates Screened Before Achieving Required Sample Size *	2207 (1595–2955)	4648 (3721–5853)	5953 (5318–6471)	3576 (3388–3788)
Post Hoc Power if 800 Recruited	49.2% (40.5–60.5%)	80.0% (71.5–87.1%)	93.1% (91.3–95.2%)	99.2% (98.9–99.4%)
PPR_min_ ***	0.433 (0.396–0.472)	0.250 (0.221–0.280)	0.903 (0.889–0.918)	0.807 (0.792–0.821)

* Assuming a refusal rate of 15% [2], with 20.3% of PWID being both eligible for in-network recruitment and susceptible (calculated from HepCEP data) and 10% of PREDICTEE candidates being recruited. Values do not account for any additional inclusion/exclusion criteria not mentioned, and thus true screening numbers may be larger. ** Refers to the susceptible Chicago PWID population. PPR_min_ was not calculated for random sample because it is sampled directly from the target population of susceptible Chicago PWID. *** Refers to the arbitrary maximally balanced target population.

## Data Availability

Our source code repository (https://github.com/sashagutfraind/vaccinetrials (accessed on 19 February 2024)) provides a copy of a spreadsheet alongside all analytical code. Publicly available synthetic data for the Chicago area can be found at https://zenodo.org/record/21714#.YshzX-zMLmo (accessed on 19 February 2024) [32]. Original survey data were licensed to the authors by third-party investigators. Readers may request access and approval will be considered on a case-by-case basis.

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
