# Peer review of "Reducing Sample Size While Improving Equity in Vaccine Clinical Trials: A Machine Learning-Based Recruitment Methodology with Application to Improving Trials of Hepatitis C Virus Vaccines in People Who Inject Drugs"

_healthcare, 2024, doi:10.3390/healthcare12060644_

Round 1

Reviewer 1 Report

Comments and Suggestions for Authors

The authors present a predictive model that balances incidence and population representation with the goal of minimizing the number of participants in clinical trials.

The work is coherent and the method presented is useful for the optimal selection of participants. The article is well written.

However, to analyse the model and analytical code in detail it is necessary to get access to the github repository that is unfortunately not working (wrong url?).

Furthermore, the supplementary material just contains the figures of the main manuscript in high resolution is that intended?

Author Response

From Reviewer: 1

The authors present a predictive model that balances incidence and population representation with the goal of minimizing the number of participants in clinical trials.

The work is coherent and the method presented is useful for the optimal selection of participants. The article is well written.

However, to analyse the model and analytical code in detail it is necessary to get access to the github repository that is unfortunately not working (wrong url?).

Response: The GitHub repository’s visibility was mistakenly blocked – it is now publicly accessible at the URL given in the paper (https://github.com/sashagutfraind/vaccinetrials/)

Furthermore, the supplementary material just contains the figures of the main manuscript in high resolution is that intended?

Response: We sincerely apologize and are perplexed by the missing appendix document. We are including the document with the resubmission.

Reviewer 2 Report

Comments and Suggestions for Authors

This study examines how a predictive model based on a behavioral model of a specific population can streamline clinical trials of vaccines, which usually require large sample sizes. This study is commendable in that it employs an agent-based model for behavioral prediction, which successfully avoids the fatal flaw of classical infection models, i.e., divergence from reality, while simultaneously validating the model's validity by comparing it with accurate epidemiological data. On the other hand, I would like to confirm the following points in utilizing this system.

1) For this system to be fully implemented in society, it is necessary to compare the cost of model design with the cost of recruiting subjects using conventional methods. As the authors also point out, a reasonable amount of detailed data on the target area may be necessary to improve the accuracy of the simulation.

2) Related to the above, it is necessary to build an API that can automatically obtain the data necessary for simulation from existing databases maintained by the government to be used as a general-purpose tool.

Author Response

From Reviewer: 2

This study examines how a predictive model based on a behavioral model of a specific population can streamline clinical trials of vaccines, which usually require large sample sizes. This study is commendable in that it employs an agent-based model for behavioral prediction, which successfully avoids the fatal flaw of classical infection models, i.e., divergence from reality, while simultaneously validating the model's validity by comparing it with accurate epidemiological data. On the other hand, I would like to confirm the following points in utilizing this system.

1) For this system to be fully implemented in society, it is necessary to compare the cost of model design with the cost of recruiting subjects using conventional methods. As the authors also point out, a reasonable amount of detailed data on the target area may be necessary to improve the accuracy of the simulation.

Response: Thank you for your valuable feedback. We agree that there may be costs related to model design and data management with PREDICTEE that we have not accounted for, and further research is needed regarding the cost-effectiveness of PREDICTEE compared to traditional recruitment methods. We have removed the cost comparison paragraph in our discussion, as we believe this is beyond the scope of the current study, and added the following paragraph to the limitations section to address this concern:

Section 4.2, lines 454-461

“Finally, more research is needed on the cost effectiveness of PREDICTEE in comparison to traditional recruitment methods. Although we anticipate that cost savings associated with recruiting fewer subjects would outweigh any additional screening costs with PREDICTEE, other costs would also have to be considered such as predictive model training, algorithm implementation, and personnel costs. This would require further studies on specific real-world PREDICTEE parameters and how they compare to traditional recruitment, such as total clinical trial duration, required volume of pre-trial data, and ease of identifying high-risk PWID.”

2) Related to the above, it is necessary to build an API that can automatically obtain the data necessary for simulation from existing databases maintained by the government to be used as a general-purpose tool.

Response: Thank you for your insightful comment. We agree that an API capable of accessing publicly available PWID or HCV data would be ideal and could greatly improve the applicability of PREDICTEE. We have added additional language in our discussion to make this point, stated below. Nonetheless, we believe that trial sites attempting to conduct a vaccine trial would already have sufficient local data to build a model for PWID to enable PREDICTEE recruitment.

Section 4.3, lines 475-477

“Moreover the adoption of PREDICTEE could be accelerated by sharing of PWID and HCV epidemiological data via an application programming interface (API).”

Reviewer 3 Report

Comments and Suggestions for Authors

This paper introduces a machine learning-based approach designed to expedite the clinical trial process and enhance its efficiency by minimizing the number of trial participants and optimizing their selection. It employs a survival analysis-based PREDICTEE strategy for recruiting, which results in a balanced demographic representation. The authors have conducted rigorous experiments, and the results seem persuasive. The paper can be accepted provided the following comments are addressed:
- Would not type-1 and type-2 errors be more suitable metrics than PPR for this issue? An examination of PPR's effectiveness is required.
- The literature review is somewhat insufficient and needs to be more comprehensive, incorporating the latest advancements.
- Please discuss the impact and limitations of proposed method.

Comments on the Quality of English Language

None

Author Response

From Reviewer: 3

This paper introduces a machine learning-based approach designed to expedite the clinical trial process and enhance its efficiency by minimizing the number of trial participants and optimizing their selection. It employs a survival analysis-based PREDICTEE strategy for recruiting, which results in a balanced demographic representation. The authors have conducted rigorous experiments, and the results seem persuasive. The paper can be accepted provided the following comments are addressed:

- Would not type-1 and type-2 errors be more suitable metrics than PPR for this issue? An examination of PPR's effectiveness is required.

Response: Thank you for your valuable feedback. We had also initially thought of using type-1 and type-2 errors as metrics for PREDICTEE efficacy; however, we were unable to do so due to the complex nature of balancing both demographic representation and HCV incidence for each candidate. Thus, we opted to measure these outcomes separately, reporting both the PPR, as a measure of demographic representation, and cohort incidence. We also explore model accuracy in our supplementary materials. To better support the use of PPR in this context, we have added the following language to our methods section and cited additional sources which validate the use of PPR for measuring external validity in clinical trials.

Section 2.3.1, lines 306-308

“…a metric that evaluates demographic representation and has been widely used in clinical trial settings for assessing adequate enrollment of demographic subgroups [50-54]”

- The literature review is somewhat insufficient and needs to be more comprehensive, incorporating the latest advancements.

Response: We have expanded our literature review to be more comprehensive by discussing recent literature relating to artificial intelligence and machine learning in clinical trials, adding the following language:

Section 1, lines 85-94

“This study contributes to a growing body of literature on the implications of artificial intelligence (AI) and machine learning in clinical trials. Recent reviews by Ismail et al. [30] and Harrer et al. [31] describe how AI/ML may be leveraged to optimize recruitment to reduce trial failures, optimize patient composition, and reduce time and costs associated with conducting an RCT. We believe PREDICTEE offers one such ML-based solution to these goals. This study also complements a recent study by Oikonomou et al. [26] which describes a ML- based phenomapping strategy capable of maximizing RCT enrollment efficiency that is similar to what we propose in our PREDICTEE methodology. However, we offer an alternative method of optimizing the clinical trial cohort and apply it to a simulated vaccine clinical trial.”

- Please discuss the impact and limitations of proposed method.

Response: We discuss the impact and limitations of PREDICTEE in our discussion, and a new section heading (4.1 Implications) has been added to emphasize our study’s impact. We have also expanded on what we believe to be the most significant limitations of PREDICTEE in the real-world setting, being possible prediction error and the necessity for pre-existing longitudinal data, and added an additional section on the necessity for further research on the cost effectiveness of PREDICTEE compared to traditional recruitment methods.

Round 2

Reviewer 1 Report

Comments and Suggestions for Authors

The authors have addressed my criticisms, the article can be published